# Profiles of Sleep Duration and Psychological Distress among Puerto Ricans Living in the United States: A Cross-Sectional Survey and Latent Class Analysis

**DOI:** 10.3390/ijerph19116363

**Published:** 2022-05-24

**Authors:** Kevin Villalobos, Francisco A. Montiel Ishino, Faustine Williams

**Affiliations:** Division of Intramural Research, National Institute on Minority Health and Health Disparities, National Institutes of Health, Rockville, MD 20852, USA; francisco.montielishino@nih.gov (F.A.M.I.); faustine.williams@nih.gov (F.W.)

**Keywords:** mental health, sleep duration, minority health, social determinants of health, health disparities

## Abstract

Sleep duration affects physiological functioning and mental health outcomes among Hispanics/Latinos. The limited research reports that Hispanic/Latino subpopulations like Puerto Ricans carry a disproportionate burden. To understand this burden, we identified profiles of sleep duration by psychological distress among Puerto Rican adults (N = 4443) using latent class analysis on 2010–2019 National Health Interview Survey data. The outcome of sleep was constructed from self-reports of sleep duration and difficulties falling and staying asleep. Sleep duration was categorized as short sleep (≤6 h), normal sleep (7 to 8 h), and long sleep (≥9 h). Mental health indicators included psychological distress items from the Kessler-6 scale. Health behaviors (nicotine and alcohol use), weight (calculated body mass index), food security, acculturative factors (language use), and socio-economic/socio-demographic covariates were considered to further differentiate profiles. Six profiles were identified: troubled shortest sleep (5.0% of sample) with high psychological distress; highly troubled short sleep (8.0%) with little psychological distress; some disturbed and shortened sleep (10.0%) with moderate psychological distress; undisturbed and shortened sleep (28.0%) with little psychological distress; normal/average sleep (49.0%); and long sleep (1.0%) with moderate psychological distress. While our study is among the first to identify profiles that are at the highest mental health risk due to sleep issues, the findings and approach help examine the complex disparity among Puerto Ricans to then leverage complex survey data for interventional designs.

## 1. Introduction

It is estimated that more than 20% of the United States (US) population has reported issues with sleep duration and/or sleep consistency [1]. Sleep is a vital component of physical and mental health that allows the biological mechanisms of the human body to function (e.g., immune system and metabolism restoration). Diseases associated with abnormal sleep duration, that is, short or long sleep, are cardiovascular disease, diabetes mellitus, and obesity [2]. Abnormal sleep duration has also been associated with cognitive decline and negative mental health outcomes [3,4]. Abnormal sleep duration is a public health concern in the US that potentially interacts with other disparities (e.g., health, environmental, sociocultural) and affects vulnerable populations such as Hispanics/Latinos at a much higher rate. According to the US Census, about 62 million or 19% of the population is Hispanic/Latino [5]. Hispanics/Latinos will reach approximately 111 million by 2060, making them the fastest growing minority group in the US [6]. Conversely, the amount of research on Hispanics/Latinos and sleep duration is limited. The limited research creates a gap not only in ongoing research but also restricts the conceptualization of how social determinants of health and sleep affect these ethnic subpopulations.

A population-based study using data from the National Health Interview Survey (NHIS) that examined racial differences in sleep duration reported that Hispanics, which excluded Mexican Americans, had an increased risk of short sleep duration in comparison to non-Hispanic Whites [7]. A similar study reported that sleep characteristics by race/ethnicity in the Chicago area found that Black, Hispanic, and Asian adult sleep duration was shorter when compared to non-Hispanic White participants [8]. Nevertheless, the range of Hispanic/Latino ethnicities in the US is a vast spectrum of cultures, norms, and languages. Research has found that sleep duration and sleep habits among Hispanics/Latinos is heterogeneous and varies within each ethnicity. For instance, Mexicans have been associated with better sleep outcomes in comparison to other Hispanic/Latino ethnicities in the US [9,10]. Another study that examined sleep patterns of Hispanic/Latino subpopulations from New York, Florida, Illinois, and California by actigraphy found that Mexicans, when compared to those that identified as Cuban, Dominican, South American, and Puerto Rican, had a shorter sleep duration [9]. This study also identified Puerto Ricans as having the highest mean sleep fragmentation in comparison to the other Hispanic/Latino ethnicities. Factors associated with sleep fragmentation are also critical to understanding sleep disparities among Hispanic/Latino subpopulations.

Alcantara et al. [11]. reported in a study that examined sociocultural stressors and sleep among Hispanics/Latinos (Central American, Cuban, Dominican, Mexican, Puerto Rican, and South American) that ethnic discrimination, acculturation, and psychosocial stressors were associated with adverse sleep outcomes such as insomnia, day-time sleepiness, short and long sleep duration. Among Hispanics/Latinos, the prevalence of mental health issues is unambiguously higher in the Puerto Rican community. A study that looked at Puerto Rican youth development of depression and anxiety from South Bronx, New York and San Juan, Puerto Rico found that the participants from South Bronx, regardless of socioeconomic status, were at increased risk of psychological distress development in comparison to the youth from San Juan [12]. A study by Canino et al. [13] that compared the prevalence of psychiatric issues among island Puerto Ricans, US mainland Puerto Ricans, and the US population reported that US Puerto Ricans had higher rates of anxiety and depression in comparison to island Puerto Ricans. Nonetheless, regardless of nativity (US-born versus island-born) the Puerto Rican community has been associated with poorer sleep duration and mental health distress [14].

The purpose of this study was to identify heterogenous profiles of Puerto Ricans based on sleep duration and mental health. While previous studies have analyzed the association between Puerto Ricans and mental health, our latent class analysis (LCA) provides a dynamic multidimensional approach that incorporates sleep profiles, mental health distress factors, and social determinants of health. We analyzed a national survey to fill a gap in the current literature concerning Puerto Ricans, sleep, mental health, and health disparities therein.

## 2. Materials and Methods

Latent class analysis was used on data from the 2010–2019 National Health Interview Survey (NHIS) on self-identified Puerto Rican adults aged 18 and older to identify mental health subpopulations by sleep profile (N = 4443). The NHIS uses a complex survey design, thus we used the appropriate weights, strata, and clusters to analyze the data. Data from the NHIS are publicly available, and all information pertaining to the NHIS can be located at the Centers for Disease Control and Prevention’s National Center for Health Statistics repository (https://www.cdc.gov/nchs/nhis/data-questionnaires-documentation.htm, accessed on 4 January 2022). No human participants were involved in this study, and thus our study was not reviewed by the Institution of Review Board. 

### 2.1. Patient and Public Involvement

This research was done without patient or public involvement. 

### 2.2. Latent Class Analysis

Mental health indicators were based on six manifestations of nonspecific psychological distress items within the past 30 days and a related follow up that was taken from the Kessler-6 instrument: (1) feeling sad [none of the time; a little to some of the time; most to all of the time], (2) feeling restless [none of the time; a little to some of the time; most to all of the time], (3) feeling nervous [none of the time; a little to some of the time; most to all of the time], (4) feeling hopeless [none of the time; a little to some of the time; most to all of the time], (5) everything felt like an effort [none of the time; a little to some of the time; most to all of the time], (6) how often did you feel worthless? [none of the time; a little to some of the time; most to all of the time], and (7) feelings interfere with life in past month [a lot; some; a little; none of the time]. Frequency or worries, nervousness, and anxiety [never; a few times a year; weekly to monthly] were also used as mental health indicators taken from the NHIS Quality of Life Supplement. Sleep profiles were constructed using self-reported sleep duration in hours, trouble falling asleep, trouble staying asleep, and feelings of tiredness or exhaustion. Sleep duration was categorized as short sleep (≤6 h), normal sleep (7 to 8 h), and long sleep (≥9 h). Trouble falling asleep in the past week and trouble staying asleep in past week were categorized as (1) none, (2) some, and (3) high degree. Feeling tired in the last three months [never; some days; most days; or every day] was taken from the NHIS Quality of Life Supplement.

#### 2.2.1. Covariates

Covariates were used in an auxiliary multinomial logistic regression to further assess the latent class membership of identified profiles. These covariates included sociodemographics, socioeconomic status, health behaviors, linguistic factors, and birthplace. Sociodemographics and socioeconomic status included sex (male/female), age (54 and younger/55 and above), sexual identity (heterosexual/bisexual or homosexual), level of education (less that college/one or more years of college), family income (at or below poverty/above poverty), and food security (yes/no). Health behaviors included cigarette smoking status (never smoker/current or former smoker), and body mass index (normal weight/overweight or obese). Linguistic factors included language of interview (other language/English). Birthplace was based on location of birth (continental US/US territory).

#### 2.2.2. Latent Class Analytic Plan

A comparative approach was used to determine the best class-solution model from a one to eight class solution. Several model fit indicators that included Bayesian information criterion (BIC), sample-size-adjusted-BIC (ssaBIC), and bootstrap likelihood ratio (BLRT) to assess the most parsimonious value, as well as high entropy (i.e., acceptable quality of classification) were used to assess model fit in addition to their practical and theoretical considerations [15]. An auxiliary multinomial logistic regression was used on the model selected for interpretation to examine how covariates predicted latent class membership. Analytical files are available upon request. All latent class analyses were conducted on Mplus 8.6 (Muthén & Muthén).

## 3. Results

The Puerto Rican sample was primarily female (59.2%), under the age of 55 (69.3), heterosexual (95.9%), and were either separated, divorced, or widowed (66.9%). They primary had an education level of high school or less (53.8%), were above the poverty level (69.4%), and had high food insecurity (66.2%). They were also primary overweight/obese (70.3%), current drinkers (55.6), and never smokers (62.2%). The language of interview was primary English (81.7%) and they were born in US territory (53.8%). See Table 1 for further detail.

As shown in Table 2, the sample had a mean sleep duration of 6.87 h and primarily reported having no trouble falling asleep (60.7%), no trouble staying asleep (62.3%), and feeling tired some days in the last three months (48.1%). The sample also reported never felt worried (44.6%), felt sad none of the time (66.9%), felt restless none of the time (63.3%), felt nervous none of the time (63.6%), felt hopeless none of the time (81.7%), and felt feelings interfered with life in the past month none of the time (33.3%), as well as everything felt like an effort none of the time (68.8%). See Table 2 for further detail. 

### 3.1. Latent Class Analysis Findings

The six-class solution was selected for interpretation (see Figure 1). Profiles were named based on the distal outcome of mental health distress, that is: (Class 1) troubled shortest sleep with high mental health distress; (Class 2) highly troubled short sleep with no indication of mental health distress; (Class 3) some disturbed and shortened sleep had some indications of mental health distress; (Class 4) undisturbed and shortened sleep had some indications of mental health distress; (Class 5) normal/average sleep with no indication of mental health distress; (Class 6) long sleep.

Class 1, or the troubled shortest sleep with high mental health distress profile (5.0% of sample), had the lowest mean sleep duration (5.47 h) when compared to all other classes. Class 1 had the highest conditional probabilities of a high degree of trouble falling asleep (81.7%) and feeling tired in the last three months every day (37.6%), as well as the second highest conditional probability of staying asleep (78.0%). The conditional probabilities of psychological distress were also the highest when compared to all other classes. That is, worried a few times a year (96.4%), feeling sad most to all of the time (72.1%), feeling restless most to all of the time (78.8%), feeling nervous most to all of the time (73.7%), feeling hopeless most to all of the time (57.6%), feelings interfered with life in past month a lot (60.9%), and everything felt like an effort most of the time (70.0%). 

Class 2, or the highly troubled short sleep with no indication of mental health distress profile (8.0% of sample), had the second lowest mean sleep duration (5.55 h) when compared to all other classes. Class 2 had the highest conditional probability of high degree trouble staying asleep (92.0%). This class had the second highest conditional probabilities for high degree of trouble falling asleep (80.0%), worried weekly to monthly (42.9%), felt tired in the last three months somedays (58.7%), and everything felt like an effort a little to some of the time (37.6%), and feelings interfered with life in the past month a little (32.2%).

Class 3, or the some disturbed and shortened sleep had some indications of mental health distress profile (10.0% of sample), had the third lowest mean sleep duration (6.71 h) when compared to all other classes. This class also had the second highest conditional probabilities of trouble falling asleep sometimes and trouble staying asleep sometimes (39.7% and 30.0%, respectively). Class 3 had the highest conditional probabilities for feeling sad a little to some of the time (89.4%), feeling restless a little to some of the time (81.9%), feeling nervous a little to some of the time (80.0%), feeling hopeless a little to some of the of time (77.4%), feelings interfere with life in the past month sometimes (32.8%), everything felt like an effort a little to some of the time (80.8%). 

Class 4, or the undisturbed and shortened sleep, had some indications of mental health distress profile (28% of sample) and had a mean sleep duration of 6.88 h. It had a conditional probability of 61.5% for trouble staying asleep none of the time. Class 4 had the highest conditional probabilities for feeling worried weekly to monthly and feeling tired in the last three months some days (59.5% and 68.5%, respectively). This class had the second highest conditional probabilities for trouble falling asleep none of the time (53.7%), feeling sad none of the time (62.6%), feeling restless a little to some of the time (49.5%), feeling nervous a little to some of the time (50.6%), feeling hopeless none of the time (91.1%), felt that feelings interfered with life in the past month (55.1%), and everything felt like an effort none of the time (62.4%). 

Class 5, or the normal/average sleep no indication of mental health distress profile (49% of sample), had a mean sleep duration of 7.23 h. This class had the highest conditional probabilities for trouble falling asleep none of the time (86.9%), trouble staying asleep none of the time (84.6%), never worried (79.0%), never felt tired in the last three months (60.4%), feeling sad none of the time (94.7%), feeling restless none of the time (99.2%), feeling nervous none of the time (96.5%), feeling hopeless none of the time (99.5%), feelings interfere with life in the past month none of the time (73.1%), and everything felt like an effort none of the time (97.5%). 

Class 6, or long sleep profile (1% of sample), had the highest sleep duration mean of 12.24 h when compared to all the classes. Class 6 had the highest conditional probability for feeling tired in the last three months most days (38.2%). This class had the second highest conditional probabilities for worried a few times a year, feelings interfered with life in the past month a lot, and everything felt like an effort most to all of the time (77.8%, 48.7%, and 55.5%, respectively). See Table 3 for further detail.

### 3.2. Covariates

Our covariate analysis using multinomial logistic regression revealed the following. In consideration of social and environmental determinants of health, we found that persons had increased odds (adjusted odds ratio (AOR) = 6.22, 95% confidence interval (CI) = 3.28–11.80) of having low food security in the troubled shortest sleep profile (Class 1) compared to the normal/average sleep profile (Class 5). This was also the case for highly troubled short sleep (Class 2) (AOR = 2.90, 95% CI = 1.85–4.56), some disturbed and shortened sleep (Class 3) (AOR = 3.74, 95% CI = 2.29–6.08), undisturbed and shortened sleep (Class 4) (AOR = 2.03, 95% CI = 1.33–3.08), and long sleep (Class 6) (AOR = 4.47, 95% CI = 1.29–15.50). In addition to having increased odds of having low food security, the highly troubled short sleep profile (Class 2) and some disturbed and shortened sleep profile (Class 3) had decreased odds of not being above the poverty line (AOR = 0.50, 95% CI = 0.30–0.84 and AOR = 0.45, CI = 0.28–0.70, respectively) compared to the normal/average sleep profile (Class 5). Among health behavior covariates, there was an increased likelihood of being a current or former smoker in the troubled shortest sleep profile (Class 1), highly troubled short sleep profile (Class 2), and some disturbed and shortened sleep (Class 3) (AOR = 2.64, 95% CI = 1.35–5.19, AOR = 1.94, 95% CI = 1.26–2.99, and AOR = 2.39, 95% CI = 1.53–3.71, respectively) compared to the normal/average sleep profile (Class 5). The undisturbed and shortened sleep profile (Class 4) had increased (AOR = 1.65, 95% CI = 1.14–2.37) odds of being current drinkers compared to the normal/average sleep profile (Class 5). When considering sociodemographics, the troubled shortest sleep profile (Class 1) and the long sleep profile (Class 6) had increased (AOR = 2.37, 95% CI = 1.18–4.77 and AOR = 3.90, 95% CI = 1.15–13.25) odds of being married when compared to the normal/average sleep profile (Class 5). The highly troubled short sleep profile (Class 2) and some disturbed and shortened sleep profile (Class 3) had increased odds (AOR = 2.79, 95% CI = 1.70–4.59 and AOR = 2.32, 95% CI = 1.46–3.70, respectively), of being female when compared to the normal/average sleep profile (Class 5). The long sleep profile (Class 6) had increased (AOR = 3.58, 95% CI = 1.10–11.64), odds of being 55 or older years of age when compared to normal/average sleep profile (Class 5). Persons were associated with increased odds (AOR = 2.28, 95% CI = 1.31–3.97), of English being the language of the interview in the undisturbed and shortened sleep profile compared to the normal/average sleep profile (Class 5). The some disturbed and shortened sleep profile (Class 3) had increased odds (AOR = 1.77, 95% CI = 1.17–2.70), of being born in US territory when compared to normal/average sleep profile (Class 5). See Table 4 For further detail. 

## 4. Discussion

We used latent class analysis, a person-centered approach, to identify the distinct profiles among Puerto Ricans adults (18+ years) based on sleep duration and mental health status. Mental health was assessed through the Kessler-6 psychological distress instrument and related follow up, as well as the Quality-of-Life Supplement administered by the NHIS. Our analysis yielded six profiles: two were identified as the shortest sleep, two were shorter sleep, one was normal/average sleep, and one profile was of a long sleep duration. The psychological distress identified in our analysis has the potential to serve as evidence of an ongoing mental health crisis among the Puerto Rican community. Our findings contribute to the emerging literature on sleep and mental health among Hispanics/Latinos in the US. Therefore, the identification of profiles while exploring sociodemographic, socioeconomic, and health behavior covariates and mental health indicators allow for an interpretation of findings and provide a possible intervention.

For instance, the troubled shortest sleep and highly troubled short sleep profiles had the highest conditional probabilities of high degree trouble falling asleep and trouble staying asleep in comparison to the other classes. A significant amount of literature has demonstrated that among Puerto Rican communities, sleep disparities are prevalent [9,14,16,17]. Prior studies have also demonstrated that in comparison to other Hispanic/Latino ethnic subpopulations, Puerto Ricans have the highest rate of sleep fragmentation [9]. However, unlike the highly troubled short sleep, the troubled shortest sleep profile presented the highest indications of psychological distress. Similar to our findings, Alegria et al. [18], examined the prevalence of psychiatric disorders among Latinos and found that Puerto Ricans had the highest overall prevalence rate when compared to other Latino ethnic subpopulations in the US (Mexican, Cuban, Other). Several studies have indicated that the increased rate of mental health distress among this ethnic subpopulation is due to the adverse experience they may have encountered. Examples of these experiences are the survival of natural disasters and/or displacement due to natural disasters like Hurricane Maria [19,20]. These adverse experiences could potentially be indicators of a mental health crisis the Puerto Rican community is experiencing. It is important to mention that although the highly troubled short sleep profile did not show indications of psychological distress, it had the highest conditional probability of high degree trouble staying asleep (92.0%). This class was more likely to be female, current smoker, overweight/obese, and have low food security. Although this profile was more likely to be overweight/obese, it was also more likely to have low food security. This could be an indication of the consumption of calorically dense food and negative health behaviors such as smoking being used as a stress coping mechanism. 

Class 3 or the some disturbed and shortened sleep profile had conditional probabilities of some trouble falling asleep (40.0%) and some trouble staying asleep (30.0%). This class had the highest conditional probabilities for feeling sad, restless, nervous, hopeless, and everything felt like an effort a little to some of the time. This class was more likely to be female, a current smoker, have low food security, and be born on the island of Puerto Rico. This profile was also less likely to not be below the poverty line. According to the US Census, about 43.5% of the population in Puerto Rico are in poverty [21]. That is, 1.4 million Puerto Ricans are below the poverty line when compared to Mississippi—the poorest state in the continental US [21]. Furthermore, we have to acknowledge natural disasters like Hurricane Maria that caused an estimated 90 billion dollars in damage, which ultimately widens the disparity gap among this subpopulation [22,23]. 

The undisturbed and shortened sleep and the normal/average sleep profile had similar profiles. The normal/average sleep profile had no mental health distress while the undisturbed and shortened sleep demonstrated some indications of mental health distress. This profile had high conditional probabilities for worried weekly to monthly, feeling restless a little to some of the time, feeling nervous a little to some of the time, and felt tired in the last three months on some of the days. This class was more likely to complete the survey in English, have low food security, and be a current alcohol drinker. This profile may be a subpopulation of second-generation Puerto Ricans in the continental US or that are highly acculturated due to the high likelihood of using English to complete the survey. Some studies have demonstrated that Puerto Ricans from the continental US have worse physical and mental health outcomes when compared to those from the Island [24,25,26]. Future studies should have an emphasis on distinguishing between Puerto Ricans born in the continental US, Puerto Ricans born on the island and stayed on the island, and those that were born on the island and moved to the continental US. These factors could potentially allow for acculturation to be taken into consideration, as acculturation could play an important role in mental health [27]. 

The long sleep profile or Class 6 had no trouble falling asleep or trouble staying asleep. This profile had some indications of mental health distress. It had high conditional probabilities for worried a few times a year, feeling restless most of the time, feeling hopeless most of the time, feelings interfered with life in the past month a lot, and everything felt like an effort most of the time. This profile was more likely to be married, above the age of 55, and have low food security. This profile is unique among the research of sleep and Puerto Ricans. The average sleep duration for this profile was 12.24 h. Several studies have demonstrated that older individuals are more likely to sleep less when compared to younger individuals [28]. This is the opposite of what we found among this profile; not only is this subpopulation sleeping more than the recommended sleep duration but also had signs of psychological distress. Future analysis should expand on information regarding sleep frequency and whether the recorded sleep was fragmented into several naps throughout the day or a constant sleep.

To our knowledge this study is among the first to apply a latent class analysis to a hard-to-reach subpopulation such as Puerto Ricans. This group is a particularly vulnerable subpopulation because of the many adverse experiences it has encountered. An example would be the several natural disasters that have affected the lives of many individuals. Another example of adversity among this subpopulation is the potential culture and ethnic confusion that this group experiences. Puerto Rico is a territory of the US, which ultimately makes them American, but as indicated by prior research and in this study, this subpopulation experiences extreme economic/financial hardship. Having said that, our study provides a window that allows us to see some of the issues/disparities among this subpopulation. Nevertheless, our study is not without limitations, one of which is the lack of data available on this subpopulation. Although previous studies have observed associations between mental health and Puerto Ricans, few studies, if any, have used sleep duration and quality as an outcome. Additionally, the data used for our analysis was cross-sectional and self-reported by the individual. This could be an issue when observing factors such as sleep or food insecurity due to inaccuracies, and could potentially insert bias. Furthermore, future analysis should incorporate acculturation factors, as this could be an essential aspect affecting this subpopulation. However, our latent class analysis could be used as evidence and provide valuable information about a subpopulation that is at increased risk. 

## 5. Conclusions

Our analysis identified six sleep duration and mental health profiles among Puerto Ricans using latent class analysis. Two of the identified profiles were short sleep; these profiles had the shortest amount of average sleep when compared to the other profiles. Another two identified profiles were shortened sleep, which were still considered short sleep but were closer to normal/average sleep. The last two identified profiles were normal/average sleep and long sleep. The findings indicate that the shortest sleep profile had the highest level of psychological distress, the most trouble falling asleep, and the most trouble staying asleep. We were also able to identify a profile of long sleep in people whose average sleep duration was 12.24 h and had some mental health distress. Ultimately, our analysis allowed us to identify unobservable profiles that are potentially at greatest risk. The results from this study can be used to improve mental health outcomes among subpopulations that are considered underserved/underrepresented and can aid as a template for intervention. 

## Figures and Tables

**Figure 1 ijerph-19-06363-f001:**
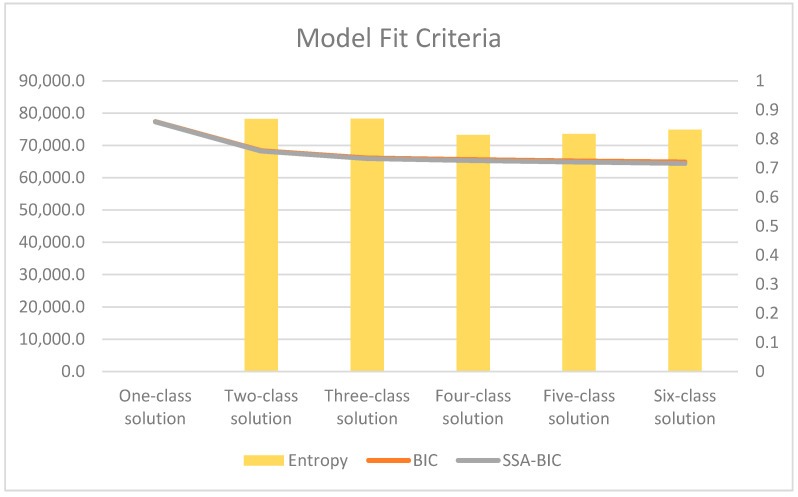
Model fit criteria for six-class solution.

**Table 1 ijerph-19-06363-t001:** Sample demographics (N = 4443).

	N	%
**Gender**		
Male	1813	40.8
Female	2630	59.2
**Age Category**		
Under 55	3079	69.3
55 & older	1364	30.7
**Sexuality**		
Heterosexual	2636	95.9
Bisexual/Homosexual	114	4.2
**Marital Status**		
Married	1467	33.1
Other (Separated, Divorced, Widowed)	2965	66.9
**Education**		
High school or less	2376	53.8
1+ years of college	2041	46.2
**Poverty level**		
Below poverty	1362	30.7
Above poverty	3081	69.4
**Food Insecurity**		
High	2597	66.2
Marginal/Low/Very low	1327	33.8
**Weight Category**		
Underweight/Normal weight	1294	29.8
Overweight/Obese	3055	70.3
**Alcohol Drinking Status**		
Former/Never drinker	1957	44.4
Current drinker	2449	55.6
**Smoke Status**		
Never	2759	62.2
Former/Current	1676	37.8
**Language of Interview**		
Spanish/English & Spanish/Other	813	18.3
English	3630	81.7
**Place of Birth**		
Continental US	2049	46.2
US territory	2390	53.8

**Table 2 ijerph-19-06363-t002:** Psychological distress (mental health) and sleep profile descriptives (N = 4443).

			Mean (SE)
**Sleep duration in hours**			6.87 (1.60)
	N	%	
**Trouble falling asleep**			
None	1875	60.7	
Some	659	21.3	
High degree	554	18.0	
**Trouble staying asleep**			
None	1922	62.3	
Some	545	17.7	
High degree	620	20.1	
**Worried**			
Never	1010	44.6	
A few times a year	540	23.8	
Weekly to monthly	714	31.5	
**Feeling tired in last three months**			
Never	764	33.8	
Some days	1089	48.1	
Most days	264	11.7	
Every day	144	6.4	
**Feeling sad**			
None of the time	2978	66.9	
A little to some of the time	1204	27.0	
Most to all of the time	270	6.1	
**Feeling restless**			
None of the time	2821	63.3	
A little to some of the time	1236	27.7	
Most to all of the time	400	9.0	
**Feeling nervous**			
None of the time	2833	63.6	
A little to some of the time	1292	29.0	
Most to all of the time	329	7.4	
**Feeling hopeless**			
None of the time	3641	81.7	
A little to some of the time	624	14.0	
Most to all of the time	191	4.3	
**Feelings interfere with life in past month**			
A lot	263	16.0	
Some	406	24.7	
A little	429	26.1	
None of the time	547	33.3	
**Everything felt like an effort**			
None of the time	3067	68.8	
A little to some of the time	966	21.7	
Most to all of the time	422	9.5	

**Table 3 ijerph-19-06363-t003:** Six-class solution conditional probabilities and mean (N = 4443).

	Class 1	Class 2	Class 3	Class 4	Class 5	Class 6
	Troubled Shortest Sleep	Highly Troubled Short Sleep	Some Disturbed and Shortened sleep	Undisturbed and Shortened Sleep	Normal/Average Sleep	Long Sleep
	N = 244	N = 343	N = 430	N = 1238	N = 2175	N = 53
	5% of Sample	8% of Sample	10% of Sample	28% of Sample	49% of Sample	1% of Sample
**Hours slept at night (Mean)**	5.47	5.55	6.71	6.88	7.23	12.24
**Trouble falling asleep**						
None	0.10	0.14	0.36	0.54	0.87	0.46
Some	0.09	0.06	0.40	0.43	0.09	0.21
High degree	0.82	0.80	0.24	0.04	0.04	0.33
**Trouble staying asleep**						
None	0.12	0.03	0.48	0.62	0.85	0.64
Some	0.10	0.05	0.30	0.33	0.10	0.16
High degree	0.78	0.92	0.22	0.06	0.06	0.21
**Worried**						
Never	0.01	0.19	0.07	0.21	0.79	0.14
A few times a year	0.96	0.39	0.64	0.19	0.04	0.78
Weekly to monthly	0.03	0.42	0.29	0.60	0.17	0.08
**Feeling tired in last three months**						
Never	0.06	0.09	0.10	0.15	0.60	0.05
Some days	0.23	0.59	0.57	0.69	0.36	0.25
Most days	0.33	0.20	0.29	0.11	0.03	0.38
Every day	0.38	0.13	0.04	0.06	0.01	0.32
**Feeling sad**						
None of the time	0.01	0.53	0.03	0.63	0.95	0.27
A little to some of the time	0.27	0.43	0.89	0.36	0.05	0.37
Most to all of the time	0.72	0.05	0.07	0.02	0.00	0.37
**Feeling restless**						
None of the time	0.04	0.34	0.09	0.46	0.99	0.35
A little to some of the time	0.18	0.47	0.82	0.50	0.01	0.24
Most to all of the time	0.79	0.19	0.10	0.05	0.00	0.41
**Feeling nervous**						
None of the time	0.06	0.46	0.09	0.47	0.97	0.28
A little to some of the time	0.20	0.42	0.80	0.51	0.04	0.37
Most to all of the time	0.74	0.12	0.11	0.02	0.00	0.35
**Feeling hopeless**						
None of the time	0.06	0.83	0.20	0.91	1.00	0.29
A little to some of the time	0.37	0.15	0.77	0.09	0.01	0.28
Most to all of the time	0.58	0.02	0.03	0.00	0.00	0.43
**Feelings interfere with life in past month**						
A lot	0.61	0.10	0.16	0.00	0.04	0.49
Some	0.27	0.29	0.41	0.15	0.08	0.20
A little	0.07	0.32	0.33	0.30	0.16	0.07
None of the time	0.06	0.29	0.11	0.55	0.73	0.25
**Everything felt like an effort**						
None of the time	0.06	0.54	0.06	0.62	0.98	0.28
A little to some of the time	0.24	0.38	0.81	0.29	0.02	0.16
Most to all of the time	0.70	0.08	0.14	0.09	0.01	0.55

Notes. Conditional probabilities that indicate likelihood within the profile are displayed in cells with white to red gradient, where white is closer to 0 and red to 1.

**Table 4 ijerph-19-06363-t004:** Multinomial logistic regression of covariates on six-class solution using Class 5 or normal/average sleep profile as reference (N = 2652).

	Troubled Shortest Sleep			Highly Troubled Short Sleep			Some Disturbed and Shortened Sleep			Undisturbed and Shortened Sleep			Long Sleep		
		95% CI			95% CI			95% CI			95% CI			95% CI	
	AOR	Lower	Upper	AOR	Lower	Upper	AOR	Lower	Upper	AOR	Lower	Upper	AOR	Lower	Upper
Female	1.48	0.86	2.57	**2.79**	**1.70**	**4.59**	**2.32**	**1.46**	**3.70**	1.37	0.95	1.99	1.00	0.38	2.61
55 or older	1.19	0.73	1.95	1.16	0.68	1.99	1.19	0.73	1.94	0.81	0.54	1.21	**3.58**	**1.10**	**11.64**
Bisexual/Homosexual	1.73	0.49	6.20	2.22	0.77	6.38	0.83	0.18	3.86	1.36	0.51	3.59	1.83	0.35	9.44
Married	**2.37**	**1.18**	**4.77**	1.03	0.66	1.61	1.27	0.81	2.00	1.14	0.80	1.63	**3.90**	**1.15**	**13.25**
1+ years of college	0.56	0.29	1.06	0.89	0.58	1.37	1.00	0.67	1.50	1.12	0.80	1.57	1.31	0.38	4.48
Above poverty line	0.76	0.42	1.38	**0.50**	**0.30**	**0.84**	**0.45**	**0.28**	**0.70**	1.03	0.68	1.58	0.78	0.32	1.91
Food insecurity	**6.22**	**3.28**	**11.80**	**2.90**	**1.85**	**4.56**	**3.74**	**2.29**	**6.08**	**2.03**	**1.33**	**3.08**	**4.47**	**1.29**	**15.50**
Current/Former smoker	**2.64**	**1.35**	**5.19**	**1.94**	**1.26**	**2.99**	**2.39**	**1.53**	**3.71**	1.18	0.81	1.73	2.10	0.75	5.87
Current alcohol drinker	0.94	0.51	1.72	1.26	0.73	2.18	1.20	0.73	1.99	**1.65**	**1.14**	**2.37**	1.59	0.58	4.39
Overweight/Obese	1.46	0.78	2.75	**1.94**	**1.15**	**3.27**	0.74	0.48	1.14	0.76	0.53	1.09	0.77	0.24	2.43
Language of Interview: English	1.24	0.74	2.07	1.96	0.91	4.22	1.04	0.61	1.80	**2.28**	**1.31**	**3.97**	1.56	0.56	4.33
Born in US territory	1.21	0.76	2.03	1.24	0.80	1.93	**1.77**	**1.17**	**2.70**	1.38	0.99	1.93	0.89	0.36	2.22

Notes. Bolded text indicates significance by 95% confidence interval and a *p* < 0.05.

## Data Availability

The authors do not have full control of the data, the data can be obtained from (https://www.cdc.gov/nchs/nhis/1997-2018.htm, accessed on 4 January 2022).

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
