# Peer review of "Profiles of Sleep Duration and Psychological Distress among Puerto Ricans Living in the United States: A Cross-Sectional Survey and Latent Class Analysis"

_ijerph, 2022, doi:10.3390/ijerph19116363_

Round 1

Reviewer 1 Report

Thank you for giving me the chance to review this interesting manuscript. This study identified sleep duration profiles by psychological distress among Puerto Ricans living in the United States. Although the topic is of interest, there are some comments need to be addressed by the authors.

Abstract:

  • It is well-written and concise.
  • It is better if you have defined the cutoff for sleep duration such as short, normal, and long sleep in the abstract as there are multiple cutoffs in the literature.

Introduction:

  • Well-written introduction related to the topic.

Materials and methods:

  • What was the original sample and the overall number of participants?
  • Add more details about the source of data and representation of the population and inclusion criteria. Also, it is important to indicate all related information regarding locations and areas of data collection. Information is needed regarding diagnosis of diseases and other outcomes.
  • In the methods, you haven’t defined sleep duration cutoffs in order to explain the results.

Results:

  • You can add a table related to covariates analysis and odds ratios. I see table 4 is enough and you can remove covariates in lines 216-248.

Discussion:

  • Without defining sleep durations explicitly, I cannot evaluate the discussion.
  • Another limitation is the lack of using objective sleep measures
  • One of the limitations is using self-reported sleep questions over one week while there are some other outcomes that can measure sleep over 4 weeks such as Pittsburgh Sleep Quality Index.

Reviewer 2 Report

The manuscript entitles "Profiles of sleep duration and psychological distress among Puerto Ricans living in the United States: A cross-sectional survey and latent class analysis" is an interesting study on the relationship between stress and sleep duration. The study identified six profiles of sleep duration in relation to stress. Furthermore, some demographic variables were used as covariates to check individual differences characterizing each profile of sleep duration. The study is important for explanation of the relationship between sleep and stress, and to develop appropriate prevention programs for people at high risk of stress and mental health problems. However, some suggestions can improve the manuscript:

  1. It is suggested to add more information about the method used in the statistical analysis. For example, authors wrote "Several model fit indicators that included Bayesian information criterion (BIC), sample-size adjusted-BIC (ssaBIC), bootstrap likelihood ratio (BLRT), and entropy were used to compare from a 1- to 8-class solution", but cut-off criteria were not given, and also references are missing. Furthermore, it is unclear whether odds ratios were adjusted in the multinomial logistic regression model. If yes, AOR should be included in table 4. If not, AOR should be implemented or replaced with OR, to avoid overestimated p-value.
  2. Table 3 needs more information, for example, it is unclear what numbers mean (please use statistical symbols to explain it).  Also, the color used in the table is not explained (description should be included in the note below table).
  3. Bold is used in table 3 but not explained. Can you use the asterisks (instead of bold) to show the p-value?

Author Response

This manuscript is a resubmission of an earlier submission. The following is a list of the peer review reports and author responses from that submission.